# Robust Optimization for Multilingual Translation with Imbalanced Data

Xian Li and Hongyu Gong

Facebook AI
{xianl, hygong}@fb.com

## Abstract

Multilingual models are parameter-efficient and especially effective in improving low-resource languages by leveraging crosslingual transfer. Despite recent advance in massive multilingual translation with ever-growing model and data, how to effectively *train* multilingual models has not been well understood. In this paper, we show that a common situation in multilingual training, data imbalance among languages, poses optimization tension between high resource and low resource languages where the found multilingual solution is often sub-optimal for low resources. We show that common training method which upsamples low resources can not robustly optimize population loss with risks of either underfitting high resource languages or overfitting low resource ones. Drawing on recent findings on the geometry of loss landscape and its effect on generalization, we propose a principled optimization algorithm, **C**urvature **A**ware **T**ask **S**caling (CATS), which adaptively rescales gradients from different tasks with a meta objective of guiding multilingual training to low-curvature neighborhoods with uniformly low loss for all languages. We ran experiments on common benchmarks (TED, WMT and OPUS-100) with varying degrees of data imbalance. CATS effectively improved multilingual optimization and as a result demonstrated consistent gains on low resources (+0.8 to +2.2 BLEU) without hurting high resources. In addition, CATS is robust to overparameterization and large batch size training, making it a promising training method for massive multilingual models that truly improve low resource languages.

## 1 Introduction

Multilingual models have received growing interest in natural language processing (NLP) [34, 41, 10, 26, 2, 19, 58]. The task of multilingual machine translation aims to have one model which can translate between multiple languages pairs, which reduces training and deployment cost by improving parameter efficiency. It presents several research questions around crosslingual transfer learning and multi-task learning (MTL) [23, 1, 2, 27, 45, 28].

Recent progress in multilingual sequence modeling, with multilingual translation as a representative application, been extending the scale of massive multilingual learning, with increasing number of languages [2, 10, 58] , the amount of data [2, 12], as well as model size [27, 13]. Despite the power-law scaling of (English-only) language modeling loss with model, data and compute [25], it has been found that multilingual models *do not* always benefit from scaling up model and data size, especially multilingual machine translation to multiple languages even after exploiting language proximity with external linguistic knowledge [12, 27, 51].

There has been limited understanding of the optimization aspects of multilingual models. Multilingual training is often implemented as monolithic with data from different languages simply combined.

35th Conference on Neural Information Processing Systems (NeurIPS 2021).

Challenges were observed when training with imbalanced data [2, 26], which is common for multilingual NLP as only a few languages are rich in training data (high resource languages) while the rest of the languages in the world has zero or low training data (low resource languages) [24]. This has been mostly treated as a "data problem" with a widely used work-around of upsampling low resource languages' data to make the data more balanced [2, 26, 10, 34, 58].

In this paper, we fill the gap by systematically study the optimization of multilingual models in the task of multilingual machine translation with Transformer architecture. Our contribution is twofold.

First, we reveal the optimization tension between high resource and low resource languages where low resources' performance often suffer. This had been overlooked in multilingual training but have important implications for achieving the goal of leveraging crosslingual transfer to improve low resources. We analyze the training objectives of multilingual models and identify an important role played by local curvature of the loss landscape, where "sharpness" causes interference among languages during optimization. This hypothesis is verified empirically, where we found optimization tension between high and low resource languages. They compete to update the loss landscape during early stage of training, with high resource ones dominating the optimization trajectory during the rest of training. Existing approaches such as upsampling low resources implicitly reduce this tension by augmenting training distribution towards more uniform. We show that this approach is not robust to different data characteristics, where it suffers from either overfitting low resources or underfitting high resources.

Second, we propose a principled training algorithm for multilingual models to mitigate such tension and effectively improve *all* languages. Our algorithm explicitly learn the weighting of different languages' gradients with a meta-objective of guiding the optimization to "flatter" neighborhoods with uniformly low loss (**C**urvature-**A**ware **T**ask **S**caling, **CATS**). Compared to static weighting implied by sampling probabilities of the data distribution, our method effectively reduced the optimization tension between high and low resource languages and improves the Pareto front of generalization. On common benchmarks of multilingual translation, CATS consistently improves low resources in various data conditions, $+0.8$ BLEU on TED (8 languages, 700K sentence pairs), $+2.2$ BLEU on WMT (10 languages, 30M sentence pairs), and $+1.3$ BLEU on OPUS-100 (100 languages, 50M sentence pairs) without sacrificing performance on high resources. Furthermore, CATS can effectively leverage model capacity, yielding better generalization in overparameterized models. The training algorithm is conceptually simple and efficient to apply to massive multilingual settings, making it a suitable approach for achieving equitable progress in NLP for every language.

## 2    Related Work

**Multilingual Learning and Multi-task Learning.**    Massive multilingual models have been gaining increasing research interest as a result of recent advancement in model and data scaling [17, 10, 23, 2, 19, 58]. However, multilingual models are often trained using the same monolithic optimization as is used in training single-language models. To deal with imbalanced training data, upsampling low resource languages (and downsampling high resource languages) was first proposed in massive multilingual machine translation in order to improve low resource performance [23, 2]. It was adopted in recent state-of-the-art multilingual pretraining [10, 34, 58].

Although recent work has looked into maximizing positive transfer across languages by learning parameter-sharing sub-networks conditional on languages [28, 31], few prior work had looked into the blackbox of multilingual optimization. Relevant efforts focus on dynamic data selection, such as adaptive scheduling [22] and MultiDDS (Differentiable Data Selection), which dynamically selects training examples based on gradient similarity with validation set [53]. Although they share the same motivation of treating multilingual training as a multi-objective optimization problem, data selection introduces additional computation and slows down training.

This work also adds to a growing interest in addressing interference in multilingual models, known as "*the curse of multilinguality*"[2, 10], which was initially hypothesized as "*capacity bottleneck*" in general multi-task learning [5]. Existing work have mainly focused on model architecture, e.g. via manually engineered or learnt parameter sharing across languages based on language proximity[61, 47, 46]. Gradient Vaccine is a recent work addressing interference from an optimization perspective by de-conflicting gradients via projection [54], a general approach MTL [60]. Although conflicting gradients are also examined in this work, they are used as evaluation metrics, where we

show that regularizing local curvature can prevent conflicting gradients from happening in the first place. Overall, we provide new understanding of interference in multilingual tasks by pointing out the optimization tension between high and low resources, which had not been studied before.

Multilingual training is an instance of multi-task learning (MTL) [9, 36, 11, 50, 44, 35]. Task balancing has been studied from both architecture and optimization perspectives [33, 6, 48, 60]. Among them, GradNorm [6] and MGDA [48] are closely related, where the GradNorm adjust gradient norms to be uniform across tasks, and MGDA adapts gradient-based multi-objective optimization to modern neural networks with efficient training.

**Sharpness of the Loss Landscape and Generalization.** Our analysis and proposed method are closely related to recent findings on the generalization of neural networks and optimization behaviors including geometric properties of loss landscape during training. It was observed that the "sharpness" of the loss landscape grows rapidly especially during the early phase of training by examining relevant metrics such as largest eigenvalues of Hessian, spectral norm of empirical Fisher Information matrix, etc. [8, 20]. Gradient alignment, measured as cosine similarity of gradients computed on different training examples, indicates the covariance of gradients, which was shown to be related to generalization performance [15, 21, 32]. Our work applies those insights from optimization literature to analyze the training dynamics of multilingual learning with the Transformer architecture. A closely-related concurrent work [14] has verified the effectiveness of regularizing loss sharpness in improving generalization, but in a single-task setting on computer vision benchmarks.

## 3 Demystifying Optimization Challenges in Multilingual Training

**Notations.** In multilingual translation task, we typically train one model with training data from a set of languages $\mathcal{N} := \{l_1, ..., l_N\}$, and measure its generalization performance on a set of evaluation languages $\mathcal{M}$. We use "language" and "task" interchangeably throughout the paper. We introduce the following notations:

- $\mathcal{D}_n := \{(x_i^{(n)}, y_i^{(n)})\}$ refers to the set of labeled examples of language $l_n$. A prominent property of multilingual translation task is the highly imbalanced data, i.e. the distribution of $|\mathcal{D}_n|$, denoted as $p_{|D_n|}$, is usually heavy-tailed, with a few high resource (HiRes) languages, and a large number of low resource (LoRes) ones [2, 55].

- Let $\boldsymbol{\theta} \in \mathbb{R}^d$ be the parameters of the multilingual model.

- $\mathcal{L}_n := \mathbb{E}_{(x,y) \sim \mathcal{D}_n}[L(\hat{y}, y)]$ as the expected loss as measured on language $n$, with token-level cross entropy loss $L(\hat{y}, y) = -\sum_i y_i^{|\mathcal{V}|} \log \hat{y}$, where $\mathcal{V}$ is the vocabulary the target language(s) which the model translates to.

- Denote $\nabla$ as gradient, $\nabla^2$ as Hessian $\mathbf{H}$, tr(.) as trace, and $\|.\|_2$ as Euclidean ($l_2$) norm.

### 3.1 Optimization Objective

Our analysis starts by taking a closer look at the training and generalization objective of multilingual models. Generalization performance is measured on a set of evaluation languages $\mathcal{M}$ weighted by their importance $w_n$, $\mathcal{L}_{\text{Multi}} := \sum_{n=1}^{|\mathcal{M}|} w_n \mathcal{L}_n(x, y|\boldsymbol{\theta})$. For multilingual translation, $w_n = \frac{1}{|\mathcal{M}|}, n = 1, ...|\mathcal{M}|$ indicating that all translation directions have equal importance. This *population loss* is usually optimized by minimizing empirical risk, i.e. a *training loss* $\hat{\mathcal{L}}_{\text{Multi}} := \min_{\boldsymbol{\theta}} \sum_{n=1}^{|\mathcal{N}|} \alpha_n \hat{\mathcal{L}}_n(x, y; \boldsymbol{\theta})$, where $\alpha_n$ corresponds to the weight of language $n$'s loss during training.

We point out that as an important optimization choice, $\boldsymbol{\alpha} := \{\alpha_n\}$ is either left implicit or chosen heuristically in existing work of multilingual training. A common choice for $\boldsymbol{\alpha}$ resembles the sampling probability of language $l_n$ in a mini-batch according to training data distribution $p_{\{|D_n|\}}^{\frac{1}{T}}$ with a temperature hyperparameter $T$. For example, $T = 1$ corresponds to sampling proportional to training data sizes $[|\mathcal{D}_1|, ...|\mathcal{D}_N|]$; $T = \infty$ corresponds to uniformly sampling across languages. For most multilingual tasks, training data is highly imbalanced with a few high resource languages accounting for majority of the training data and a long tail of low resource languages. Practitioners manually tune $T$ to make minimizing training loss $\hat{\mathcal{L}}_{\text{Multi}}$ a proxy of optimizing the population loss

$\mathcal{L}_{\text{Multi}}$. Massive multilingual translation and pretraining with real-world datasets of 100 languages usually upsample low resource languages (e.g. $T = 5$) to improve the averaged generalization performance [2, 34, 10, 58].

## 3.2 Local Curvature and Robust Multi-task Optimization

We analyze how local curvature affects optimization efficiency of multilingual training. Task interference has been observed to pose optimization challenges in general multi-task learning and continual learning, where improvement in one task hurts performance on another task [45, 48, 30, 60, 38].

In the context of training multilingual models, we define *interference* from the optimization perspective, as the change of empirical risk $\mathcal{L}_1$ as a result of an update to $\boldsymbol{\theta}$ dominated by empirical risk minimization for another language's loss $\mathcal{L}_2$. In Figure 1, we illustrate how curvature of the loss landscape affects interference similar to the analysis which has been done in continual learning [38]. After a gradient step driven by $\nabla\mathcal{L}_2$, the shared model parameters moved from $\theta_1^*$ to $\theta_2$. The corresponding changes in $\mathcal{L}_1$, i.e. $\mathcal{L}_1(\theta_2) - \mathcal{L}_1(\theta_1^*)$ is affected by the curvature.

Figure 1: **Left:** Change of one language's loss $\mathcal{L}_1$ after shared parameters being updated to $\boldsymbol{\theta_2}$ driven by $\nabla\mathcal{L}_2$ from another language is affected by the curvature of the loss landscape around previous critical point $\boldsymbol{\theta_1^*}$. **Right:** Illustration of the proposed algorithm, Curvature Aware Task Scaling (**CATS**), to learn task weighting re-scaling $\alpha_n$ such that the combined gradients will guide the optimization trajectory to low curvature region (pointed by the green arrow).

$$I(\mathcal{L}_1, \mathcal{L}_2) \triangleq \mathcal{L}_1(\theta_2) - \mathcal{L}_1(\theta_1^*) \approx (\theta_2 - \theta_1^*)^\top \nabla\mathcal{L}_1(\theta_1^*) + \frac{1}{2}\nabla^2\mathcal{L}_1(\theta_1^*)\|\theta_2 - \theta_1^*\|_2^2 \quad (1)$$

This relationship is also summarized in Eq. (1). We make an assumption that in a small neighborhood of $\boldsymbol{\theta_1^*}$, the loss surface is almost convex[7, 16]. Then we can apply a second-order Taylor expansion of $L_1(\boldsymbol{\theta_2})$ in Eq. (2), and derive the connection between $I(\mathcal{L}_1, \mathcal{L}_2)$ and the Hessian $\mathbf{H}(\boldsymbol{\theta}) = \nabla^2\mathcal{L}(\boldsymbol{\theta})$ which indicates the local curvature of loss surface.

$$\mathcal{L}_1(\theta_2) \approx \mathcal{L}_1(\theta_1^*) + (\theta_2^* - \theta_1^*)^\top \nabla\mathcal{L}_1(\theta_1^*) + \frac{1}{2}(\theta_2 - \theta_1^*)^\top \nabla^2\mathcal{L}_1(\theta_1^*)(\theta_2 - \theta_1^*) \quad (2)$$

Since $\nabla\mathcal{L}_1(\theta_1^*) \approx 0$ at critical point for $\mathcal{L}_1$, the major contributing factor to interference is local curvature, measured by the spectral norm of $\mathbf{H}(\theta_1^*)$, as well as the magnitude of parameter update $\|\theta_2 - \theta_1^*\|_2^2$. We hypothesize that the optimization tension defined above affects low resource languages more than high resource ones, assuming that low-resource tasks have lower sample complexity, and thus are more likely to reach local minima during early stage of training.

## 3.3 Meta-learn $\alpha$ with curvature regularization

Motivated by previous section's analysis, we propose a principled optimization procedure for multilingual translation with a meta objective of regularizing local curvature of the loss landscape of shared parameters $\boldsymbol{\theta}$. We explicitly *learn* the task weighting parameters $\boldsymbol{\alpha} := [\alpha_n]_{n=1,...,N}$ so as to minimize the trace norm of empirical Fisher information $tr(\boldsymbol{F}) = \mathbb{E}_{(x,\hat{y})}[\|\sum_{n=1}^{N}\alpha_n\nabla_\theta\hat{\mathcal{L}}_n\|^2]$, which is an approximation of $tr(\mathbf{H})$ as was proposed in optimization literature [20, 52]. To leverage distributed training in modern deep learning, we estimate $\boldsymbol{\alpha}$ from a mini-batch $\mathcal{B}$ with $K$ languages as is shown in Eq. (3):

---

**Algorithm 1** Curvature Aware Task Scaling (**CATS**).

---

**Input:** a mini-batch $\mathcal{B}$ with $K$ languages; number of inner loop updates $m$.
**Output:** converged multilingual model $\boldsymbol{\theta}$.

1: Initialize $\boldsymbol{\theta}, \boldsymbol{\alpha} = \mathbf{1}$ , $\boldsymbol{\lambda} = \mathbf{0}$.
2: **while** not converged **do**
3:     **for** $i$ from 1 to $m$ **do**
4:         **for** $n = 1, ..., K$ **do**
5:             Compute $g_n = \nabla_{\boldsymbol{\theta}}\hat{\mathcal{L}}_n(\mathcal{B}_n)$
6:             $\tilde{g}_n = \text{clone}(g_n)$
7:             $\tilde{\alpha}_n = \text{clone}(\alpha_n)$
8:         **end for**
9:         Update $\boldsymbol{\theta}$ with $\sum_{n=1}^{K}\tilde{\alpha}_n g_n$
10:     **end for**
11:     Compute $\nabla_{\boldsymbol{\alpha}}\hat{\mathcal{L}}_{\text{meta}_\alpha}, \nabla_{\boldsymbol{\lambda}}\hat{\mathcal{L}}_{\text{meta}_\alpha}$ according to Eq. (4) using $\tilde{g}_n$ for $\nabla_{\boldsymbol{\theta}}\hat{\mathcal{L}}_n$
12:     Update $\boldsymbol{\alpha}$ with gradient descent
13:     Update $\boldsymbol{\lambda}$ gradient ascent
14: **end while**

---

$$\min_{\alpha_1,...,\alpha_K}\|\sum_{n=1}^{K}\alpha_n\nabla_{\boldsymbol{\theta}}\hat{\mathcal{L}}_n(\boldsymbol{\theta})\|^2 \text{ s.t. } \sum_{n=1}^{K}\alpha_n = 1, \alpha_n \geq 0, \forall n \quad (3)$$

We treat solving Eq. (3) along with minimizing empirical risk $\hat{\mathcal{L}}_{\text{Multi}}$ as a multi-objective optimization problem, and optimizing the corresponding Lagrangian [4]. The overall training objective is as follows:

$$\hat{\mathcal{L}}_{\text{CATS}}(\boldsymbol{\theta}, \boldsymbol{\alpha}, \boldsymbol{\lambda}) = \sum_{n=1}^{K} \alpha_n \hat{\mathcal{L}}_n - \lambda_c (\epsilon - \|\sum_{n=1}^{K} \alpha_n \nabla_{\boldsymbol{\theta}} \hat{\mathcal{L}}_n\|^2) + \lambda_s (\sum_{n=1}^{K} \alpha_n - 1)^2 - \sum_{n=1}^{K} \lambda_n (\alpha_n - \epsilon) \quad (4)$$

We learn both model parameters $\boldsymbol{\theta}$ task weighting parameters $\boldsymbol{\alpha}$ simultaneously with bi-level optimization, where update $\boldsymbol{\theta}$ in the inner loop and update $\boldsymbol{\alpha}$ and Lagrangian multipliers $\boldsymbol{\lambda} := [\lambda_c, \lambda_s, \lambda_1, ..., \lambda_N]$ in the outer loop. We refer to proposed algorithm as CATS (Curvature-Aware Task Scaling) with the detailed training procedure described in Algorithm 1.

## 4 Experiments

### 4.1 Experimental setup

**Datasets.** We experiment on three public benchmarks of multilingual machine translation with varying characteristics of imbalanced data as is shown in Table 1. They are representative in terms of the number of languages $|\mathcal{N}|$, total volume of training data $|\mathcal{D}_n|$ measured as the number of sentence pairs in millions (M), and the entropy of data distribution $H_{|\mathcal{D}_n|}$ indicating the degree of data imbalance. For example, distributions with multiple high resource languages are covered by experiments on the TED and

| | $|\mathcal{N}|$ | $|\mathcal{D}_n|$ | $H_{|\mathcal{D}_n|}$ |
|---|---|---|---|
| TED [53] | 8 | 0.8 | 0.71 |
| WMT[34] | 10 | 31 | 0.21 |
| OPUS-100[62] | 92 | 55 | 4.2 |

Table 1: Description of datasets and characteristics of data imbalance in three representative multilingual translation benchmarks.

OPUS100 datasets, while experiments on WMT dataset cover a unique scenario of extremely skewed distribution with one high resource language and a long tail of low resource ones, which is not uncommon in real world applications. Additional details and statistics of the datasets are provided in Appendix A.

**Models.** We use the Transformer architecture, and the same model size and configuration as were used in corresponding baselines. For experiments on TED, the model is a 6-layer encoder-decoder, 512 hidden dimension, 1024 FFN dimension, and 4 attention heads [53]. For experiments on OPUS-100, the model is the Transformer-base configuration as is used in [62]. We use Transformer-base for WMT experiments. We use the same preprocessed data by the MultiDDS baseline authors [53], and followed the same procedure to preprocess OPUS-100 data released by the baseline [62]. All models are trained with the same compute budget for comparison. We provide detailed training hyperparameters in Appendix B.

**Baselines.** We compare to strong baselines used in state-of-the-art multilingual translation and relevant approaches in generic multi-task learning:

- Proportional sampling. This is a straightforward yet common approach used in practice by training on combined training data from all languages [62], which corresponds to $\alpha_n = p^{\frac{1}{T}}, T = 1$.

- Upsampling low resources $T = 5$. This has been adopted in state-of-the-art multilingual transformers as is discussed in Section 3.1.

- MultiDDS[53]: A recently proposed method to balance training losses in multilingual training. It learns to select training examples among different languages based on gradient similarity with validation set. Although it does not address the optimization challenges studied in this work, it shares the same interest of improving generalization performance and utilizing training examples in a dynamic and adaptive fashion.

- GradNorm[6]: We also compare to a closely related approach proposed for general multi-task learning. The key difference is that the objective in GradNorm is to rescale each task's gradients to be closer to the average gradients norm (i.e. only based on $\nabla L_i$), while the objective in our approach is to minimize the second-order derivative $\nabla^2 L_i$ which corresponds to the curvature of the loss landscape.

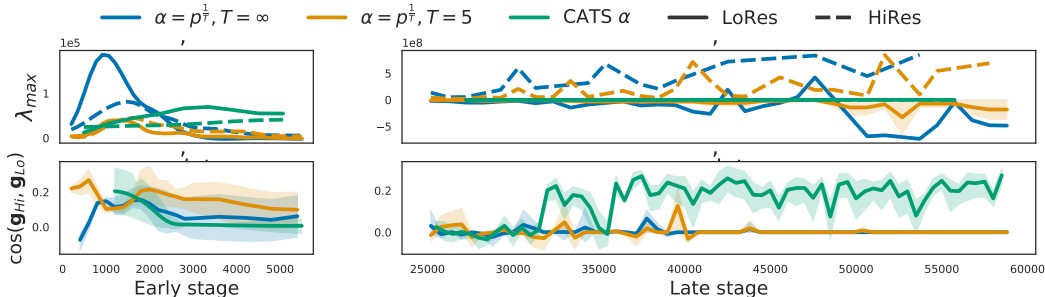

Figure 2: Local curvature measured by top eigenvalue (**Top**) and gradient direction similarity (**Bottom**) of multilingual training with high-resource (HiRes) and low resource (LoRes) languages measured on TED corpus. To control for other factors such as language proximity, both high resource (Russian) and low resource (Belarusian) are chosen to be from the same language family. We compare the proposed optimization method CATS with existing approaches of setting $\alpha_n$ based on empirical distribution of training examples, i.e. $\alpha_n \propto p_{|D_n|}$, and upsampling with a temperature hyperparameter $T$. $T = \infty$ corresponds to equal weighting (uniform sampling). In the beginning of training (**Left**), HiRes and LoRes competes to increase the sharpness the loss landscape, with HiRes dominating the optimization trajectory during the rest of the training (**Right**) and their gradients are almost orthogonal. Our proposed method (CATS $\alpha$) effectively reduced local curvature and improved gradients alignment.

**Evaluation of Optimization.** For the analysis of loss landscape, we look into the top eigenvalue ($\lambda_{max}$) of $\mathbf{H}$ computed from examples in a mini-batch with the power iteration method described in [59] for each language. To evaluate optimization efficiency, we analyze gradient similarity using the same metrics introduced in [60]. At each training step, with $\boldsymbol{g}_1 = \nabla\mathcal{L}_1$, $\boldsymbol{g}_2 = \nabla\mathcal{L}_2$ computed from training examples of two languages, we measure the following metrics as indicators of gradient similarity:

- Gradient direction similarity (alignment): $\cos\phi(\boldsymbol{g}_1, \boldsymbol{g}_2)$ where $\phi$ is the angle between $\boldsymbol{g}_1$ and $\boldsymbol{g}_2$.
- Gradient magnitude ($\mathcal{L}_2$ norm) similarity: $\gamma(\boldsymbol{g}_1, \boldsymbol{g}_2) = \frac{2\|\boldsymbol{g}_1\|_2\|\boldsymbol{g}_2\|_2}{\|\boldsymbol{g}_1\|_2^2 + \|\boldsymbol{g}_2\|_2^2}$.

**Evaluation of Generalization.** We verify whether improved optimization leads to better generalization. We report both token-level loss (negative log-likelihood, NLL↓) and BLEU scores (↑) on hold-out datasets. we choose the best checkpoint by validation perplexity and only use the single best model without ensembling.

We conduct most experiments in multilingual one-to-many task as it is a more challenging task than many-to-one, and represents the core optimization challenges underlying many-to-many task. Additional experiments with many-to-one are provided in Appendix C.

## 4.2 Results

### 4.2.1 Robust Optimization

**Abrasive Gradients between HiRes and LoRes.** First, we illustrate the optimization tension between high resource languages and low resource languages (HiLo). We examine abrasive gradients in Figure 3, which shows a fine-grained view of gradients similarity in terms of both direction (left) and magnitude (right) for different parameters in the Transformer architecture: encoder self attention (E.SA), encoder feed-

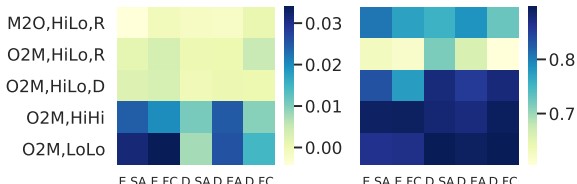

Figure 3: Gradients similarity of different Transformer parameters ($x$-axis) in common multilingual translation tasks ($y$-axis), measured as gradients direction similarity (**Left**), and gradients norm similarity (**Right**).

forward (E.FC), decoder self-attention (D.SA), decoder encoder attention (D.EA), and decoder feed-forward (D.FC). We can see that gradients are more similar when training with balanced data, i.e. HiHi and LoLo, compared to HiLo for two fundamental multilingual translation tasks one-to-many (O2M) and many-to-one (M2O), indicating that the amount of training data is less a cause of the problem compared to the distribution of training data across languages. Furthermore, contrary to

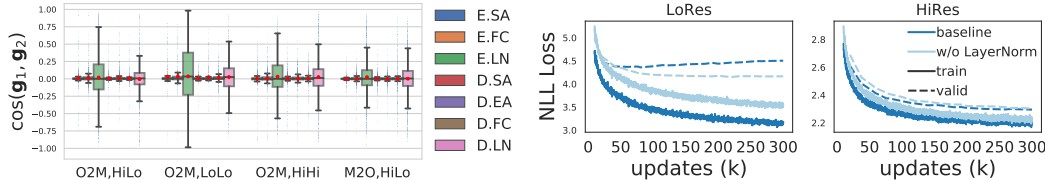

(a) Gradient alignment ($y$-axis) for Transformer parameters across common multilingual translation tasks ($x$-axis).

(b) Stable training without LayerNorm parameters and mitigating LoRes overfitting.

Figure 4: Gradients alignment between HiRes and LoRes have high variance for Layer normalization parameters (LayerNorm) in both encoder (E.LN) and decoder (D.LN), which exacerbate overfitting of LoRes. CATS allows stable training without LayerNorm parameters and reduces generalization gap for LoRes.

common heuristics which use language family information to determine which languages share parameters [12], language proximity, i.e. related (R) vs. diverse (D), has a smaller effect on gradient similarity compared to data imbalance.

Next, we look into optimization tension through the lens of curvature. Figure 2 (top) plots the top eigenvalues $\lambda_{max}$ when training a multilingual model in the HiLo setting. We can see that with uniform sampling (equal weighting, $T = \infty$), LoRes updates the shared parameter space in the direction towards higher curvature during the early stage of training, while HiRes dominates the optimization trajectory (with the LoRes having negative top eigenvalue) for the rest of the training. We found that common heuristics of $T = 5$ reduces the early-stage sharpness although the loss landscape is still dominated by HiRes. In contrast, the proposed optimization approach (CATS $\alpha_n$) mitigated the abrasive optimization and increased gradients alignment as is illustrated Figure 2 (bottom).

**Training without layer normalization.** We take a closer look at gradients alignment for different types of parameters in the Transformer architecture. We found that gradient alignments for layer normalization parameters (LayerNorm, LN) are much noisier than other parameters as is shown in Figure 4a. Extensive work has provided empirical and theoretical results of LayerNorm be being critical for training stability of Transformers[3, 57, 56]. Our work is the first to show that it is a double-edged sword in multilingual training and exacerbates the tension between HiRes and LoRes, which competes to set the gain and bias parameters in LayerNorm, where the final weights highly depend on training data distributions. However, simply removing the gain and bias parameters causes training instability[32]. In Figure 4b, we show that with CATS optimization in place, Layer-Norm can be removed yet stable training can still be achieved without further adjusting other hyperparameters such as decreasing learning rate, etc. As a result, the generalization gap (difference between training and validation loss) for low resource is greatly reduced possibly due to that the learnt Layer-Norm parameters would otherwise be biased towards HiRes.

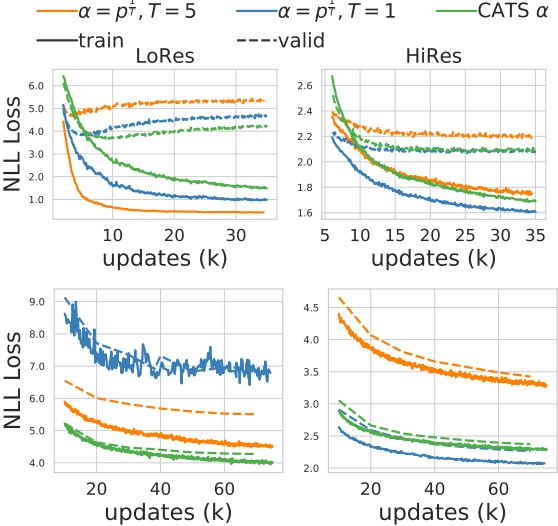

Figure 5: Train and validation loss (token-level negative log-likelihood, NLL ↓) for low resource (**Left**) and high resource (**Right**) from the same multilingual model. We can see that addressing data imbalance with the temperature hyperparameter $T$ is not robust to changing data distribution $p_{|\mathcal{D}_n|}$: TED (**Top**) and WMT (**Bottom**). For highly imbalanced WMT dataset, the common practice of upsampling low resource ($T = 5$) improves LoRes but at the cost of underfitting HiRes, while it leads to LoRes overfitting for less imbalanced dataset (TED). Datasets characteristics are described in Table 1.

| | Related | | | | | | | | | Diverse | | | | | | | | |
|---|---|---|---|---|---|---|---|---|---|---|---|---|---|---|---|---|---|---|
| | LoRes | | | | HiRes | | | | Avg. | | LoRes | | | | HiRes | | | | Avg. |
| | aze | bel | slk | glg | por | rus | ces | tur | All | Lo | bos | mar | hin | mkd | ell | bul | fra | kor | All | Lo |
| $|\mathcal{D}_n|$ (K) | 5.9 | 4.5 | 61.5 | 10.0 | 185 | 208 | 103 | 182 | | | 5.6 | 9.8 | 18.8 | 25.3 | 134 | 174 | 192 | 205 | | |
| $T=\infty$ | 5.4 | 9.6 | 24.6 | 22.5 | 39.3 | 19.9 | 22.2 | 15.9 | 19.9 | 15.5 | 12.7 | 10.8 | 14.8 | 22.7 | 30.3 | 31.3 | 37.2 | 16.9 | 22.1 | 15.3 |
| $T=5$ | 5.5 | 10.4 | 24.2 | 22.3 | 39.0 | 19.6 | 22.5 | 16.0 | 19.9 | 15.6 | 13.4 | 10.4 | 14.7 | 23.8 | 30.4 | 32.3 | 37.8 | 17.6 | 22.5 | 15.6 |
| $T=1$ | 5.4 | 11.2 | 23.2 | 22.6 | 39.3 | 20.1 | 21.6 | 16.7 | 20.0 | 15.6 | 13.0 | 11.4 | 15.0 | 23.3 | 32.9 | 34.0 | 40.5 | 18.8 | 23.5 | 15.7 |
| GradNorm | 5.3 | 9.8 | 24.5 | 22.6 | 38.9 | 20.1 | 21.9 | 15.4 | 19.8 | 15.6 | 13.7 | 11.4 | 14.3 | 23.1 | 29.0 | 30.2 | 35.1 | 16.3 | 21.6 | 15.6 |
| MultiDDS | 6.6 | 12.4 | 20.6 | 21.7 | 33.5 | 15.3 | 17 | 11.6 | 17.3 | 15.3 | 14.0 | 4.8 | 15.7 | 21.4 | 25.7 | 27.8 | 29.6 | 7.0 | 18.2 | 14.0 |
| CATS $\alpha$ | 5.4 | 11.7 | 24.7 | 23.2 | 40.6 | 20.6 | 22.6 | 17.0 | **20.7\*** | **16.3\*** | 12.1 | 11.9 | 15.6 | 24.7 | 33.3 | 35.7 | 41.8 | 19.3 | **24.3\*** | **16.1\*** |

Table 2: Comparison of CATS with common methods used in multilingual training. We evaluate on the 8-language TED benchmark with related (top) and diverse (bottom) languages which help verify performance while controlling optimization difficulty due to language proximity. We compare to static weighting with manually tuned temperatures $T = 1, 5, 100(\infty)$, and dynamic weighting such as GradNorm [6] and MultiDDS [53]. Results are BLEU scores on test sets per language and the average BLEU score (↑) across all languages (All) and low resource languages (Lo). Multilingual training which achieves the **best average** BLEU is in bold and the strongest baseline approach is annotated with underscore. * indicates the improvements are statistically significant with $p < 0.05$.

**Robust to different training data distributions $p_{|\mathcal{D}_n|}$.** We show that the improved optimization is reflected in generalization performance. In Figure 5, we report the training and validation loss of HiRes and LoRes languages throughout training on two representative benchmarks: TED (top) and WMT (bottom) with different degrees of data imbalance. We can see that the optimal temperature hyperparameters $T$ vary given the distinct distributions of training data $p_{|\mathcal{D}_n|}$ in these two benchmarks. For example, on WMT dataset where the training data distribution across languages is more skewed, upsampling LoRes ($T = 5, \infty$) is beneficial, as was observed in other large-scale multilingual

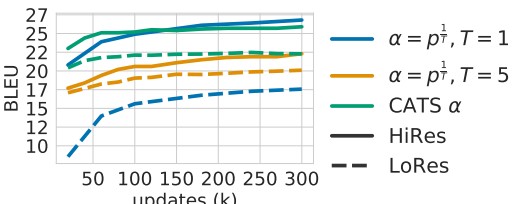

Figure 6: CATS is very effective in highly imbalanced datasets (WMT), where common sampling approaches $T = 1, 5$ sacrifice either low resources (LoRes) or high resources (HiRes). CATS significantly improves generalization on LoRes while demonstrating better sample efficiency.

datasets [2, 10, 34]. However, $T = 5$ easily leads to overfitting for LoRes on a more balanced dataset such as the TED dataset. Compared to existing approaches of using a hyperparameter $T$, CATS is more robust as a principled way to address varying distributions of training data. It does not need manual tuning of a hyperparameter $T$ nor suffer from overfitting or underfitting. On both benchmarks, it consistently improves LoRes without sacrificing HiRes.

### 4.2.2 Generalization

We report translation quality on three representative benchmarks whose detailed characteristics are described in Table 1.

**TED.** Table 2 summarizes translation quality improvement on the TED corpus. To control for language similarity, we evaluate on the 8-language

| | LoRes | | | | | | | MidRes | | HiRes | Avg. | |
|---|---|---|---|---|---|---|---|---|---|---|---|---|
| | kk | vi | tr | ja | nl | it | ar | ro | hi | de | All | Lo |
| $|\mathcal{D}_n|$ (M) | 0.1 | 0.1 | 0.2 | 0.2 | 0.2 | 0.3 | 0.3 | 0.6 | 0.8 | 27.8 | | |
| $T=1$ | 1.1 | 27.3 | 13.5 | 13.3 | 28.6 | 25.2 | 14.0 | 29.8 | 14.1 | **26.8** | 19.4 | 17.6 |
| $T=5$ | 1.2 | 31.3 | 15.5 | 14.9 | 30.9 | 30 | 16.9 | 31.1 | 14.7 | 22.3 | 20.9 | 20.1 |
| CATS $\alpha$ | **2.0** | **33.1** | **19.2** | **16.7** | **33.3** | **32.1** | **19.6** | **35.4** | **18.0** | 25.9 | **23.5\*** | **22.3\*** |

Table 3: Detailed performance on the WMT benchmark. Compared to the strongest baselines ($T = 1, 5$), CATS drastically improves low and mid resource languages. Multilingual training which achieves the **best** BLEU is in bold and the best baseline is annotated with underscore.

benchmark with both *Related* (4 LoRes and 4 HiRes from the same language family) and *Diverse* (4 LoRes and 4 HiRes without shared linguistic properties) settings as is used in [53]. We found simple mixing ($T = 1$) is a very strong baseline and complex training methods such as GradNorm

and MultiDDS do not generalize well (TED is a smaller dataset and more prone to overfitting). CATS achieves $+0.4 \sim 0.7$ BLEU on average for LoRes and $+0.7 \sim 0.8$ BLEU overall.

**WMT.** On a larger and more imbalanced 10-language WMT datset with 30.5M sentence pairs, we found regularizing local curvature with meta learnt $\alpha$ (CATS) is more beneficial. CATS improved low and mid resource languages across the board as is shown in Table 3. Compared to the strongest baseline ($T = 5$), CATS achieved $+2.2$ average BLEU for low resource languages and $+2.6$ BLEU across all languages. Furthermore, CATS is more sample efficient as is shown in Figure 6.

**OPUS-100.** Finally, we test the scalability of CATS in massive multilingual setting using the OPUS-100 benchmark[62]. Results are summarized in Table 4. Consistent with previous results on OPUS-100 [62, 61] as well as other massive multilingual model in [2], upsampling low resource with $T = 5$ improves BLEU scores for low resource but at the cost of accuracy for high resource ones. The trade-off is the opposite for proportional sampling ($T = 1$). CATS achieves the *the best of both worlds*, especially $+1.3$ BLEU on low resource, $+1.4$ BLEU on mid resource, and $+1.0$ BLEU on high resource compared to the strong baseline of $T = 5$. We also compare to a recent state-of-the-art model, conditional language-specific routing (CLSR) [61], which adds additional language specific parameters (157M parameters in total compared to 110M parameters in our experiments). By improving optimization without increasing model capacity, CATS still outperforms CLSR on low resource by $+0.9$ BLEU.

|  | LoRes | MidRes | HiRes | All |
|---|---|---|---|---|
| $|\mathcal{D}_n|$ | < 100K | | ≥ 1M | |
| $|\mathcal{N}|$ | 18 | 29 | 45 | |
| $T = 1$ | 16.4 | 22.8 | **22.1** | 21.2 |
| $T = 5$ | 26.8 | 24.6 | 18.9 | 22.3 |
| CATS $\alpha$ | **28.1*** | **26.0*** | 19.9 | **23.4*** |
| CLSR[61] | 27.2 | 26.3 | 21.7 | 23.3 |

Table 4: Performance on OPUS-100. CATS can easily apply to training at the scale of $\sim 100$ languages and improves low resources.

## 5 Analysis

**Robust to overparameterization.** Scaling up model size has been of central interest in recent development of massive multilingual models such as GShard and Switch Transformer with trillions of parameters [27, 13]. Training overparameterized models a shorter amount of time have shown to be more efficient than training a smaller model for longer time [29, 39, 25]. Therefore, we are interested in understanding CATS' performance in training overparameterized models. Figure 7 plots change in generalization (measured by BLEU score difference) as we increase model capacity. CATS can effectively benefit from larger model capacity, especially in the overaparameterized regime (300M parameters for TED dataset) where performance begins to degradate with standard training.

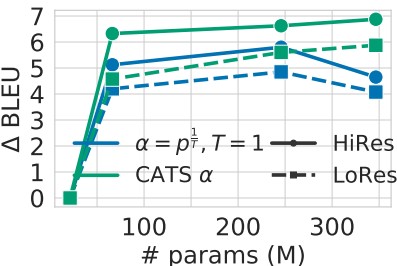

Figure 7: CATS improves generalization in overparameterized models, while standard approach suffers from overfitting.

**Performance with large batch size.** Large batch size is an effective way to speed up training without sacrificing performance[18, 49]. However, heterogeneous training data do not necessarily benefit from large batch size training[37]. This is a challenge for multilingual training as is shown in Figure 8, where increasing batch size hurts generalization for LoRes with a common practice of upsampling ($T = 5$). This is likely due to LoRes is prone to overfitting (illustrated in Figure 5) and larger batch size exacerbates it. The same batch size without upsampling leads to improved generalization ($T = 1$). In comparison, CATS can effectively benefit from larger batch size due to its adaptive rescaling of gradients.

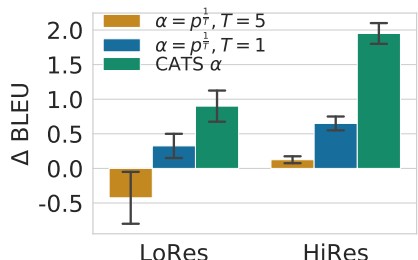

Figure 8: CATS is robust in large batch size training ($4\times$ batch size from 33K to 131K tokens).

**Ablation layer normalization.** To further understand the effect from removing layer normalization which mitigates LoRes overfitting as was shown in Section 4.2.1, we provide an ablation study on its role when combined with CATS. We ran experiments on TED Diverse dataset with the strongest baseline ($T = 1$). In Table 5, we can see that CATS brings additional $+0.5$ BLEU to low resources *on top of* the $+0.3$ BLEU improvement from removing layer normalization parameters (-LN).

| | LoRes | | | | HiRes | | | | Avg. | |
|---|---|---|---|---|---|---|---|---|---|---|
| | bos | mar | hin | mkd | ell | bul | fra | kor | All | Lo |
| Baseline | 24.3 | 10.7 | 22.9 | 33.5 | 38.0 | 39.1 | 40.5 | 19.1 | 28.5 | 22.8 |
| -LN only | 24.4 | 10.6 | 23.7 | 33.9 | 38.4 | 39.5 | 40.9 | 19.5 | 28.9 | 23.1 |
| CATS | 24.6 | 11.3 | 24.2 | 34.1 | 39.1 | 40.1 | 41.2 | 19.6 | 29.3 | 23.6 |

Table 5: Ablation study of the effectiveness of CATS when combined with removing layer normalization parameters (-LN).

## 6 Conclusion

In this work, we look under the hood of monolithic optimization of multilingual models. We unveil optimization challenges arising from imbalanced training data, where low resource languages have been sub-optimally optimized. We proposed a principled optimization algorithm for multilingual training with adaptive gradient rescaling of different languages where the scaling is learnt with a meta-objective of guiding the optimization to solutions with low local curvature.

We evaluated the proposed method on three representative benchmarks (TED, WMT and OPUS100) which cover a wide range of imbalanced data distributions commonly seen in real word multilingual datasets. Compared to existing methods of simply mixing the data or manually augmenting the data distribution to be more "balanced" with a temperature hyperparameter, the proposed training method demonstrates robust optimization and consistently improves generalization for low resource languages. Further analysis shows that the proposed approach is suitable for the realistic training settings of large-scale multilingual learning such as overparameterized models, large batch size, highly imbalanced data, as well as large number of languages etc., paving the way for advancing massive multilingual models which truly benefit low resource languages.

**Broader Impact.** Recent progress in NLP enabled by scaling up model size and data is widening the gap of technology equity between high resource languages (such as English) and low resource languages. Multilingual model is a promising approach to close this gap. However, current *language agnostic* multilingual models does not effectively improve low resources for various reasons to be understood. Our investigation in optimization is one step towards building *truly inclusive multilingual models*.

This work has several limitations which we hope to extend in future work. We did not run experiments on translation between non-English directions including zero-shot. The absolute size of the overparameterized model in our analysis is relatively small compared to the state-of-the-art of model scaling of billions of parameters. As an language generation application, machine translation model could produce unsafe output or hallucination.

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
