# A Dataset Statistics

For TED experiments, we use the same preprocessed data[1] provided by [53] using the same train, valid, and test split as in [43]. The data volumes for related and diverse language groups are summarized in Table 6.

Languages are grouped into families based on the their similarity. For TED Related dataset, we have four language families: Turkic family with Azerbaijani and Turkish, Slavic family with Belarusian and Russian, Romanance family with Glacian and Portuguese, and Czech-Slovak family with Slovak and Czech. As for the Diverse dataset, the languages are grouped into five families: Indo-iranian family with Hindi and Marathi, Slavic family with Macedonian, Bosnian and Bulgarian, Korean family with Korean, Hellenic family with Greek, and Romance family with French.

Table 6: Data Statistics of TED Datasets

|         | Code | Language   | Train  | Dev  | Test |
|---------|------|------------|--------|------|------|
| Related | aze  | Azerbaijani | 5.94k  | 671  | 903  |
|         | bel  | Belarusian | 4.51k  | 248  | 664  |
|         | glg  | Glacian    | 10.0k  | 682  | 1007 |
|         | slk  | Slovak     | 61.5k  | 2271 | 2445 |
|         | tur  | Turkish    | 182k   | 4045 | 5029 |
|         | rus  | Russian    | 208k   | 4814 | 5483 |
|         | por  | Portuguese | 185k   | 4035 | 4855 |
|         | ces  | Czech      | 103k   | 3462 | 3831 |
| Diverse | bos  | Bosnian    | 5.64k  | 474  | 463  |
|         | mar  | Marathi    | 9.84k  | 767  | 1090 |
|         | hin  | Hindi      | 18.79k | 854  | 1243 |
|         | mkd  | Macedonian | 25.33k | 640  | 438  |
|         | ell  | Greek      | 134k   | 3344 | 4433 |
|         | bul  | Bulgarian  | 174k   | 4082 | 5060 |
|         | fra  | French     | 192k   | 4320 | 4866 |
|         | kor  | Korean     | 205k   | 4441 | 5637 |

The source of WMT dataset is public parallel corpora from previous WMT shared task. We use the same preprocessed data as was used in several multilingual translation tasks [34]. We selected 10 languages with highly imbalanced data distribution and diverse linguistic features from 5 language families: (1) Arabic; (2) Kazakh and Turkish; (3) Vietnamese; (4) German, Hindi, Italian, Dutch, Romanian; (5) Japanese.

Table 7: Data Statistics of WMT Dataset

| Code | Language | Size | Code | Language   | Size  |
|------|----------|------|------|------------|-------|
| kk   | Kazakh   | 91k  | vi   | Vietnamese | 133k  |
| tr   | Turkish  | 207k | ja   | Japanese   | 223k  |
| nl   | Dutch    | 237k | it   | Italian    | 250k  |
| ro   | Romanian | 608k | hi   | Hindi      | 1.56M |
| ar   | Arabic   | 250k | de   | German     | 28M   |

# B Implementation Details

**Training.** We applied the same regularization for both baseline and the proposed method, such as label smoothing 0.1, dropout probability 0.3 for TED corpus and 0.1 for WMT and OPUS-100. For training, we use FP16 training, Adam optimizer with learning rate $lr = 0.0015$, $\beta_1 = 0.9$, $\beta_2 = 0.98$, inverse square root learning rate schedule with 4000 updates warm-up. For experiments

---

[1]The authors of [53] provided the downloadable data at `https://drive.google.com/file/d/1xNlfgLK55SbNocQh7YpDcFUYymfVNEii/view?usp=sharing`

Table 8: Performance on TED corpus with four low resource (LoRes) and four high resource (HiRes). To control for language proximity, we evaluate on the 8-language benchmark with both related and diverse languages in multilingual many-to-one (M2O) translation. We compare to static weighting with commonly used temperature hyperparameters $T = 1, 5$, and dynamic weighting such as MultiDDS [53]. Results are BLEU scores on test sets per language and the average BLEU score ($\uparrow$) across all languages (All) and low resource languages (Lo). Multilingual training which achieves the best BLEU is in **bold** and the strongest baseline approach is annotated with underscore.

| | | LoRes | | | | HiRes | | | Avg. | |
|---|---|---|---|---|---|---|---|---|---|---|
| Related | aze | bel | slk | glg | por | rus | ces | tur | All | Lo |
| $T = 5$ | 5.5 | 10.4 | 24.2 | 22.3 | 39.0 | 19.6 | 22.5 | 16.0 | 19.9 | 15.6 |
| $T = 1$ | 5.4 | 11.2 | 23.2 | 22.6 | 39.3 | 20.1 | 21.6 | 16.7 | 20.0 | 15.6 |
| MultiDDS | 6.6 | 12.4 | 20.6 | 21.7 | 33.5 | 15.3 | 17 | 11.6 | 17.3 | 15.3 |
| CATS $\alpha$ | 5.7 | 11.7 | 24.1 | 23.3 | 39.7 | 20.5 | 21.9 | 16.8 | **20.5** | **16.2** |
| Diverse | bos | mar | hin | mkd | ell | bul | fra | kor | All | Lo |
| $T = 5$ | 21.4 | 8.9 | 19.6 | 30.4 | 37.3 | 38.4 | 39.8 | 18.9 | 26.8 | 20.1 |
| $T = 1$ | 24.3 | 10.7 | 22.9 | 33.5 | 38 | 39.1 | 40.5 | 19.1 | 28.5 | 22.9 |
| MultiDDS | 25.3 | 10.6 | 22.9 | 32.1 | 35.3 | 35.8 | 37.3 | 16.8 | 27.0 | 22.7 |
| CATS $\alpha$ | 24.6 | 11.3 | 24.2 | 34.1 | 39.1 | 40.1 | 41.2 | 19.6 | **29.3** | **23.6** |

with LayerNorm, we use the PreNorm setup which has been shown to be more robust [40, 56]. We use maximum 50k updates with batch size of 131k tokens for the TED corpus, maximum 300k updates with batch size of 262k tokens for the WMT dataset, and maximum 500k updates with batch size of 262k tokens for the OPUS-100 corpus. Each experiment were run with 32 Nvidia V100 GPUs (32GB).

**CATS.** $\alpha$ is initialized with $\frac{1}{N}$, where $N$ is the number of languages. We use standard SGD for updating $\alpha$ with learning rate chosen from $\{0.1, 0.2\}$ based on validation loss. $\lambda$ are initialized with 0 and updated with gradient ascent. We set the number of languages per batch $K = 4$ for TED and WMT experiments and $K = 8$ for OPUS-100 experiments. Due to the high curvature at early stage of training, we update $\alpha$ at higher frequency with $m = 10, 100$ for the first 8k and 16k updates and $m = 1000$ for the rest of the training. We also tested $m = 1$ for the early stage and found the optimization trajectories do not change much.

**Evaluation.** We use the best checkpoint (without ensembling) chosen by validation loss. We use beam search with beam size 5 and length penalty 1.0 for decoding. We report SacreBLEU[42].

## C   Additional Experiments

We report experiments on many-to-one (M2O) translation on the TED corpus in Table 8.