# OpenReview forum: "Robust Optimization for Multilingual Translation with Imbalanced Data"
_NeurIPS.cc/2021/Conference — NeurIPS 2021 Poster_

### Official Review · Reviewer_V2N9 · 2021-07-12

**Rating:** 7
**Confidence:** 3

**Summary:**

This paper theoretically motivates and then implements a curvature-based, online, per-language learning rate adjustment to improve the performance of low-resource languages in multilingual machine translation (MT) models. Experiments in three scenarios varying in multilinguality and data size show that the technique works consistently, outperforming a widely-adopted temperature-based sampling baseline. Through a number of analysis and ablation experiments, they are able to provide insight into how and why the optimization technique works.

**Ethical Concerns:**

None.

**Limitations And Societal Impact:**

This is handled nicely in the appropriate section.

**Main Review:**

This paper takes a step toward solving an important problem (balancing languages in multilingual MT) using a well-motivated technique influenced by recent advances in the optimization literature. It has a very good balance of clear motivation, theoretical background, empirical results and analysis. The method may be too complex to see wide adoption without a clever or clear open source implementation (the baseline, after all, is very very easy to build), but the ideas and the connections to the optimization literature are likely to be influential regardless. The paper definitely has more than enough technical content to warrant a conference paper, and is bordering on a journal paper. I learned a lot and earmarked many other papers from the substantial bibliography to follow up on.

In terms of weaknesses, the paper is very dense. In particular, the analysis sections require a lot of effort to understand what was done and why. I spent much more time than I would have liked staring at legends, captions (sometimes multiple captions per figure!), and cross-referencing with the text to try to figure out what’s going on. Both halves of Figure 5 are particularly confusing. However, this time was generally worth the effort.

Small concerns:

The paper is littered with small disfluencies (particularly confusion of singular and plural nouns and corresponding agreement), but I found it did not hinder my understanding.

I found the definition in Section 3 of p_{|D_n|} confusing, but its intended meaning quickly became clear after seeing its use in the (familiar) temperature sampling baseline.

In the first paragraph of section 4.2.1 when Figure 5b is referenced, I’m 95% sure Figure 5a is what was intended.

Also in 4.2.1, I found the phrase “data volume” ambiguous and confusing. I guess you mean the total amount of training data for all languages? As opposed to the amount of data for any one language?

In general, the relationship between CATS (the proposed method) and layer norm is not clear in the main text. Figure 5b and the “Training without layer normalization” paragraph, together with the “Ablation layer normalization” paragraph seem to suggest that layer normalization is always switched off for CATS throughout all of the paper’s main results, but this is never explicitly stated in the main text. Likewise, in Figure 5b, it isn’t clear what the baseline system is (is it CATS or T=5 or T=1?). Presumably, the w/o layer norm line is just the baseline (whatever it is) without layer norm.

**Time Spent Reviewing:**

4

---

> ### Author Response · Authors · 2021-08-10
> **Author Response**
>
> We thank the reviewer for the thoughtful comments and very valuable feedback! We are encouraged that the reviewer recognized that this work is a timely contribution tackling an important yet under-studied problem in the field. We appreciate that reviewers found our experiments comprehensive and sound, acknowledging the technical strength of ours work especially on bridging recent advances in deep learning optimization with concrete applications such as multilingual machine translation with equitable improvement for low resources.
>
> We are also grateful for a very thorough review and detailed suggestions. We will incorporate the feedback and suggestions in the revised version:
> - We will improve the organization of the paper to be less dense and easier to read.
> - Thanks for your feedback about $p_{|D_n|}$ and “data volume”. We will make them more clear.
> - You are totally correct that the reference to Figure 5b is a typo. We will fix it.
> - The feedback on adding more clarity of layer norm is very valuable. Both models being compared in Figure 5b are with CATS, as simply removing the layer norm parameters (gain and bias) causes training instability in standard optimization of Transformers. Figure 5b illustrate that layer norm parameters could be a double-edged sword in multilingual models, that is despited the well-studied benefit of layer normalization in general Transformer training, it is susceptible to high resource and low resource languages competing to update the shared gain and bias parameters. "w/o layer norm" removed such tension by removing the layer norm parameters, yet it can still train stably with CATS optimization.

---

### Official Review · Reviewer_kfHY · 2021-07-20

**Rating:** 6
**Confidence:** 4

**Summary:**

This paper studies the problem of robustness optimization in multilingual machine translation models. In response to this problem, the author empirically found that in the loss optimization of multilingual machine translation models under unbalanced data scenarios, "sharpness" of local curvature in the loss landscape causes interference among languages, which caused instability in training. Inspired by the robust optimization work of Multi-task Learning, the paper proposes the Curvature Aware Task Scaling (CATS) optimization algorithm, which alleviates the phenomenon that different translation directions competitive update the loss landscape during the early stage of training in multi-language translation training, while high resource ones dominating the optimization trajectory during the rest of training.

**Limitations And Societal Impact:**

Yes

**Main Review:**

Strength:
1. The proposed CATS optimization method has achieved better results than the multi-objective optimization methods: GradNorm and MultiDDS.
2. The paper has done a very good analysis and visualization of the "sharpness" of local curvature in the loss landscape for the multilingual machine translation model in the unbalanced data scenario, making the research problem very intuitively presented.
3. This work also conducts experimental exploration on more practical scenarios: overparameterization and large batch size, which makes the proposed method more practicable.

Weakness:
1. The unbalanced data scenario has not been properly explored by experiments. Under what circumstances can it be counted as an unbalanced data scenario, and what is the data ratio? Therefore, the experiments should not pay more attention to one given setting like TED, WMT, etc., but should construct unbalanced scenarios of different ratios by sampling data in one setting like WMT to verify this important issue.
2. There is a lack of a reasonable ablation study on the upsampling parameter T, so we cannot confirm whether the oversampling overfit phenomenon will occur, and to what extent will the upsampling reach.
3. Some baselines are missing in the experimental comparison, such as 1) giving different weights to the loss of unbalanced translation pairs so that in the later stages of training, there will be no situation where rich-resource pairs dominate the training loss; 2) the use of low-resource language pairs further finetune the multilingual model and use the method like R3F to maintain the generalization ability of the model.
4. In some low-resource language translations from 1.2->2.0, although the improvement of 0.8 can be claimed, it is insignificant in a practical sense.

Missing References:
1. Aghajanyan, Armen, et al. "Better Fine-Tuning by Reducing Representational Collapse." International Conference on Learning Representations. 2020.

**Time Spent Reviewing:**

6

---

> ### Author Response · Authors · 2021-08-10
> **Author Response**
>
> We thank the reviewer for providing concrete feedback to help us improve the paper! We are glad that the strength of this paper is recognized by the reviewer, and the reviewer put effort into suggesting further improvements. We found there may be some minor confusions which stem from lack of rich context in our writing. We would like to to provide some clarification with more background to address possible misunderstanding in the weakness section:
>
> - On “construct unbalanced scenarios of different ratios by sampling data in one setting like WMT “: we are in agreement with the motivation of studying unbalanced scenarios with different ratios”, which is exactly why we experimented with three representative benchmarks (TED, WMT, and OPUS100), each of which cover a unique scenarios of data imbalance with different ratios (among languages) while all together they achieve a good coverage of major scenarios, such as
>   - TED: data imbalance between high and low resource but more homogeneous within, i.e. multiple and even number of high resource and low resources. Furthermore, the linguistic similarity is further controlled (by Related and Diverse settings).
>   - WMT: highly skewed distribution one or two high resources but a long tail of low resources.  This scenario is not uncommon in real world applications, e.g. English being the highest resource language (e.g. pretrained language models) dwarfing all other languages.
>   - OPUS100: a distribution with larger number of languages (100) and sizable number of high, mid, and low resource languages respectively. This was the only real public dataset with such a wide coverage of languages for multilingual translation task.
>
>   More importantly, these datasets reflect the true resource distribution in the real world and the real challenges in training multilingual models. We recognized that constructing those scenarios by sampling data from one setting (e.g. WMT) as is suggested has the pros of enabling direct comparisons across different degrees of data imbalance within one dataset, but it has the cons that the resulting distributions being artificial and not reflecting the wide range of scenarios and challenges in real world, and we think the cons outweigh the pros.  We will improve our writing to provide this background of our experiment design rationale.
> - Regarding “There is a lack of a reasonable ablation study on the upsampling parameter T”, we would like to clarify that the proposed approach does not use updampling, therefore we did not see the need to do ablation on sampling temperature. However, we did provide a comparison of upsampling with T=5 in Figure 4 which already showed overfitting. Originally, we also compared T=$\infty$ in Figure 4 but removed it since we found it crowds the figure. We will add it back in the Appendix.
> - For the “missing baselines”: 1) “giving different weights to the loss of unbalanced translation pairs “ is actually achieved by the GradNorm baseline. 2) “use of low-resource language pairs further finetune the multilingual model” do you think it can correspond to upsampling low resource, e .g. T=5 baseline we included? As is discussed below, R3F is a method for improving finetuning of pretrained models. We found it is addressing a different problem with possible different underlying causes of the optimization problem.
> - Thank you for suggesting additional references. “"Better Fine-Tuning by Reducing Representational Collapse." proposed a robust finetuning method to address the training instability problem of finetuning large pretrained models. We can see the very loose connection in terms of high-level goal, i.e. more robust optimization. Although we found this work provides insights from a remote problem, we think this work is not immediately relevant to the problem being studied in this work, where imbalanced data across languages casts challenges to the current practice of monolithic optimization. The second “missing reference” (Lin et al. 2021) first appeared in public (arxiv) on May 19th (i.e. right before NeurIPS submission deadline). It improves multilingual translation from an architectural perspective (orthogonal to optimization which is the focus of this work). We will include it in the related work section.
> - We agree with you. We did not claim the improvement from 1.2 to 2.0 BLEU to be significant, and it does not affect the overall conclusion from the experiment.

---

### Official Review · Reviewer_SuVh · 2021-07-20

**Rating:** 8
**Confidence:** 5

**Summary:**

This work looks at the optimization of multilingual NMT models. Typically, the data across different language pairs is highly skewed and prior work either upsamples low-resource languages such that it's uniform or uses a temperature (T=5) to upsample lowRes and downsample HiRes. This work proposes a new optimization algorithm called CATS which encourages solutions with low local curvature. The proposed technique has been shown to be effective when compared to 3 temperature-based baselines (T=1,5,infinity) and two dynamic methods multiDDS and GradNorm. Experiments are performed on 3 public datasets: TED, WMT and OPUS-100. Robustness of the technique to overparameterization and to large batches has been studied.

**Ethical Concerns:**

None.

**Limitations And Societal Impact:**

Yes.

**Main Review:**

Originality:
- The paper approaches the optimization problem from a novel standpoint and establishes that low local curvature affects optimization. Then it proposes an algorithm that jointly also learns weights \alpha which regularizes local curvature of the loss landscape.

Clarity:
- For the most part, the paper is clearly written. However, there are a lot of minor mistakes. The figure numbers are mixed up and the paper could have been more polished.

Positives:
- The paper proposes a novel optimization algorithm called CATS which alleviates the lowres/highres interference problem in multilingual NMT.
- Results on 3 standard benchmarks against many strong baselines show that the proposed approach leads to improves across lores, midres and hires language-pairs.

Negatives:
- The paper should have had a dedicated section on the complexity of the proposed approach. This has largely been ignored. How much extra time does the additional optimization add to training? How stable is the proposed approach?
- There are a lot of presentation mistakes which could have been easily avoided and a bit more care in organizing figures and tables better could have helped the reader.

Minor comments:
- L74: I think the idea of upsampling low-resource languages was already proposed in [20].
- L125: $N$ should be \mathcal{N}?
- L249-251: "Contrary to popular belief..." can you please cite prior work that states that language similarity has this effect?
- L241 Figure 5b shows... Do you mean Figure 3?
- Figure 4 I believe the legend should say T=infinity and not T=1.
- L279 "empirical an" --> "empirical and"
- Section 4, why was the T=infinity baseline not evaluated for WMT and OPUS-100 corpora?
- L367 "principle optimization.." --> "principled.."?


**Time Spent Reviewing:**

3

---

> ### Author Response · Authors · 2021-08-10
> **Author Response**
>
> We thank the reviewer for their thorough reviews and insightful feedback! We appreciate that the reviewer values the comprehensive experiments we conducted while studying an important problem filling the gap in the current research landscape. We are also encouraged that the originality and impact of this work is recognized.
>
> We will act on reviewer's feedback to improve the organization and writing. For example, we will add a detailed discussion of the complexity and sensitivity of the proposed approach. We found there was negligible difference in terms of training speed and memory cost since we do not need to perform the additional computation from CATS at each update. We will provide a detailed training speed comparison. Regarding training stability, we listed major hyperparameters we experimented with which are important for training stability and we will expand those with more details. We will also improve on the minor comments the reviewer pointed out (thank you again for your thorough review!):
>
> - We will fix the typos and formatting errors (e.g. L125, L241, L279 and L367).
> - L74: You’re totally right. This approach was already being used in [20] (before its importance being emphasized in [2]). We will add the reference accordingly.
> - L249-251: Good point. We will add the missing citations.
> - This is a very good question. We included the $T=\infty$ baseline in TED because it was shown to be a strong baseline for this dataset, as was observed in MultiDDS as well. The reason is likely due to the training data *within* high resources and low resources are more uniform while the data imbalance exist *between* high and low resources. WMT and OPUS100 datasets cover two additional scenarios of data distribution  with a common property of more skewed distributions than TED. Therefore, uniform sampling was not a strong baseline in those scenarios. We will add a clarification on this.

---

### Official Review · Reviewer_aPhj · 2021-07-25

**Rating:** 6
**Confidence:** 5

**Summary:**

Paper presents a curvature aware bi-level optimization algorithm applied to multilingual/multi-task machine translation. The proposed algorithm is designed to encourage the loss is reduced along the directions of low-curvature (with the implicit inductive bias of flat minimas being better at generalization). Akin to "Sharpness-Aware Minimization (Foret et al. 2020)" but in a multi-task environment. Experiments on three different datasets tests the effectiveness of the proposed method (called Curvature Aware Task Scaling - $CATS$). Results show clear trend that the proposed method is highly effective at improving the quality of low-resource languages. Further analysis provided to understand the tradeoffs, sensitivity and dynamics of the algorithm.

- Making use of curvature information during the optimization of multi-task models for translation is novel and expected to be impactful in the long run (while the paper is not demonstrating convincing empirical results, it is showing the potential of using higher order signals during optimization, which I found to be an important contribution).

**Main Review:**

At high level paper proposes to learn to weight/scale the gradients of different languages/tasks in a bi-level optimization setup. The signal used for scaling the gradients is coming from the curvature (flatness) of the loss landscape, where the inductive bias is to guide the optimization towards flatter regions of the loss landscape.

Authors found that optimization tension or the tug-of-war between high and low resource languages is due to the observation that they compete to update the loss landscape during early stage of training, with high resource ones dominating the optimization trajectory during the rest of training, and connect this to the existing methods (like upsampling low-resource languages).

Paper provides a new perspective on the problem of interference in multilingual models which I found novel.

**Questions**
- What is the number of inner optimization steps `m`
- Figure 1(right) is not clear, how do the colored arrows relate to the (left)? Perhaps add some additional comments to the caption and the main text.
- Figure 3 (left) the range of similarities are quite narrow, $\pm 0.01$, I'm a bit reluctant to make claims about gradient similarities (in terms of direction) given the range of the Figure, color-coding is misleading. But I agree that the available signal is somewhat supporting the argument "gradients are more similar when training with balanced data".
- What is the inner optimization step rule, `Adam`, `SGD` or else?
- What is the value of the epsilon in $Eq.4$ ?
- In Section 3 you mention that you *only* report one-to-many results, but in $Fig.3,5$ you mention many-to-one. I couldn't get the motivation, could you please clarify?
- Section 4.2.1 refers to $Fig.5$ but i guess it should be citing to $Fig.3$ right?
- What is the method you used to compute the top-eigenvalues of the Hessian? $Fig.2$
- You have a section detailing the use of layer-norm and its impact on the gradient alignment. What is the architecture you are using in the first place? Pre-norm Transformer or post-norm?
- When you use $CATS\alpha$ how do we choose the sampling ratios? Also in the appendix you mention that you mix K=4 (TED&WMT) and K=8 (OPUS100), what is the rationale behind these, and did you measure the sensitivity of $CATS\alpha$ for K.
- If we use T=5 and $CATS\alpha$ would that yield us the best adaptive schedule? Could you please elaborate on this?
- When you are plotting the NLL ($Fig. 4,5$), are you using $Eq.4$ ? What is the value we are seeing in the plots? (Asking because there is almost $1.0$ diff between different methods vs $CATS\alpha$)
- In WMT experiments, only German is added to the mix and is considered to be high-resource language. While it is the highest resource language in the mix (28M examples), it is still dwarfed by languages in the WMT (Czech, Chinese, French, Spanish ...) in terms of the amounts of data. I'm curious why you chose to add only 1 high resource language in the mix? Considering the gap between temperature based sampling and $CATS\alpha$ in $Table.3$ I suspect the ranking would change if you were to add more high resource languages. Given that the community is mixing high and low resource languages in a more uniform way (ie. by not including only one high resource language but as many as available), do you think the choice of languages is introducing a selection bias and skewing the results? I'm very much okay with the framing and main contribution as boosting the quality for low-resource languages btw.
- Experimental results does not support your argument on "without hurting high resources (line 18, line 64)". The only realistic high-resource setting in all your experiments (TED, WMT, OPUS) is the WMT English->German direction, and with the experimental results you present in $Table 3$ (T=1 26.8 BLEU vs $CATS\alpha$ 25.9 BLEU) we cannot support the argument (also in $Table.4$ if we "assume" language pairs that have 1M training examples being "high-resource", final results are contradicting with the argument 22.1 vs 19.9 BLEU). Please revise accordingly (I think low-resource boost with this principled approach is still a valuable contribution, but we should be cautious about not overclaiming).
- Baselines: I was puzzled by the motivation behind the selected baselines (Table 2). GradNorm is a method that has never been applied to the domain of interest here. If you would like to compare your method with a solid baseline exploiting the gradient directionality during optimization, it would be better to compare it against "Gradient Vaccine". Similarly, MultiDDS is making use of the validation set information, which is making it hard to compare since its inductive bias is quite different. I think the closest approach to your proposed method is Foret et al. 2020 "Sharpness-Aware Minimization for Efficiently Improving Generalization", which is also not applied to multilingual translation (but along the lines of using GradNorm as a baselines). I'd really like to see a section discussing what are we trying to measure by the selected "baselines".


**About the background and related work:**
-  2nd paragraph of your intro: [1] and [2] are not multilingual translation papers. While they are related work, they do not support the argument you are trying to make in the paragraph (which grounds the motivation). Please either remove the unrelated papers from the motivation (which will help the reader to locate the relevant information) or change the framing of the paragraph to include NLP in general.
- Again in the 2nd paragraph, papers [10, 24, 46] are not providing the evidence you are using to motivate the manuscript. The benefits on some dimensions start to diminish, but some others keep improving with more scale and data (eg. the quality on (extremely) low-resource languages start to diverge and regress - but this only supports the argument on scaling model, ). Rather than motivating it along the line of deficiency of one particular branch in the multilingual NMT (data, scale, # languages), you can still motivate your work since optimization is a canonical branch encapsulating all. In short, I don't think you need to motivate your work which improves the optimization angle by showing the deficiency of other branches, but rather by showing the lack of compelling approaches that address optimization in particular (which the 3rd paragraph does nicely).
- Please credit [34] for the section 3.2 where both the motivational figure, equations and the narrative are very much alike with aligned flow, and perhaps point the reader to [34] for more details.
- On Interference, or "the curse of multilinguality", paper [8] rebrands the interference problem, which was studied and introduced by [2] with "capacity bottleneck". It is not clear that the problem is exclusively related to the "multilinguality", as it could be observed under several other scenarios like multi-domain, multi-view, multi-class problems (in fact the capacity bottleneck is introduced in the seminal paper by Caruana
 on multi-task learning where the abstraction is over tasks, not languages). I believe this will make the framing of the paper more generic and applicable beyond "multilingual" translation domain. If you don't want to abstract it further, please either be more specific under multilingual translation context, or apply proper credit assignment if you would like to highlight when and under which context it was introduced.
- One thing that made me frown a few times while reading the paper was the way paper uses the previous work (citation rationale). Not all the citations are really supporting the argument the authors are trying to make, they sometimes appear to be remotely related or not related at all (we can of course bundle all of them under NLP applications if we wanted to). Please do another pass with this in mind and clean the citations. I hope the authors will improve this point.

**Typos**
- line 190: "pubic" --> public
- Sub-captions of Figure 5 are hard to decouple between (a) and (b). Could you please use some more spacing?
- line 264: temperate -> temperature


**Time Spent Reviewing:**

8 hours

---

> ### Author Response · Authors · 2021-08-10
> **Author Response**
>
> We thank the reviewer for their thorough reviews and insightful feedback! We are encouraged that the novelty and unique contribution of this work is recognized. We are especially grateful to comprehensive suggestion on improving the presentation of related work. Below we provide clarifications to answer the questions in detail. We will improve the writing by incorporating the reviewer's suggestions as well as fixing typos and formatting errors.
> - We provided detailed hyperparameters in the Appendix. Basically a smaller value of m will impose higher degrees of regularization and slow down training more. Due to the high curvature at the early stage of training, we update α at higher frequency with m = 10, 100 for the first 8k and 16k updates  and m = 1000 for the rest of the training. We also tested m = 1 for the early stage and found the  optimization trajectories do not change much.
> - That's great feedback and suggestion. The arrows in Figure 1 (right)  illustrate the change of gradient moving towards loss surfaces with lower curvature (green color), as Figure 1 (left) shows that it imposes less interference (comparing green vs. red loss surfaces). We will add a comment to make this more clear.
> - This is also provided in the Appendix. We use standard SGD for updating $\alpha$ with learning rate chosen from {0.1, 0.2} based on validation loss. Lagrangian multipliers $\lambda$ are initialized with  0 and updated with gradient ascent.
> - The $\epsilon$ in Eq. 4 play the role of an upperbound or lowerbound in the reformulated Lagrangian optimization and its value is closely tied to the term which we want to optimize (in this case the combined gradients norm and non-negative constraints in Eq. 3). Therefore, we chose $\epsilon$ empirically to be boundary value of the target term.
> - One-to-many (O2M) translation has been shown to be more challenging than many-to-one (M2O) from empirical results in the literature and the cause has been hypothesized to be interference.  We analyzed abrasive gradients for both. Since we did not find M2O poses additional challenges, we only focused on O2M for further evaluations (e.g. on generalization in Sec 4.4.4).
> - Correct. L241 should be Figure 3 instead of Figure 5b. We will fix this. Thanks for your careful reading!
> - We used the power iteration method proposed in [1]. We will add more details and the reference.
> - We were using pre-norm in the Transformer which had been shown to be more robust than post-norm [2, 3, 4].
> - We do not adjust the sampling ratio with CATS training, i.e. we just sample based on the true data distribution across languages. The number of languages (K) is chosen based on the batch size limit. For TED experiments, we did not observe training trajectory was sensitive to K=4 or K=8.
> - As is mentioned above, training with CATS eliminates the need to adjust sampling ratio with a temperature parameters. As we have shown in the paper, CATS is a more robust and principled approach to deal with imbalanced data in multilingual training.
> - NLL in Figure 4 and 5 refers to token-level negative log-likelihood and is defined in L115. It corresponds to the empirical risk $\hat{L}_n$ in Eq. 4.
> - This is a good question. Our experiments on TED, WMT and OPUS100 aim to cover a wide range of data imbalance. Distributions with multiple high resources are covered by the TED and OPUS100 experiment, while the WMT experiment covers a unique scenario of extremely skewed distribution. This scenario is not uncommon in real world applications, e.g. English being the highest resource language (e.g. pretrained language models) dwarfing all other languages. However, it reveals the drawback of existing practice of manually augmenting the data distribution to be more “balanced” with a temperature parameter $T$, that it is sub-optimal and not robust to varying distributions.
> - Thanks for your feedback. The gaps in high resources (> 1M, 22.1 vs. 19.9) were not statistically significant. We will improve the phrasing to be more clear.
> - This is a very insightful suggestion. The concurrent work from Foret et al. 2020 is indeed the closest besides GradNorm.  We chose GradNorm as a baseline since both work belong to the category of rescaling gradients (norm only) without explicitly projecting gradients (changing both direction and norm), which is the approach taken by Gradient Vaccine. The gradient projection operation is more costly than simply rescaling the norm. We will follow the suggestion to add a discussion on the rationale of choosing baselines.
>
>
> [1] Yao, Zhewei, Amir Gholami, Kurt Keutzer, and Michael W. Mahoney. "Hessian-based analysis of large batch training and robustness to adversaries." In Proceedings of the 32nd International Conference on Neural Information Processing Systems, pp. 4954-4964. 2018.
>
> [2] Xiong, Ruibin, Yunchang Yang, Di He, Kai Zheng, Shuxin Zheng, Chen Xing, Huishuai Zhang, Yanyan Lan, Liwei Wang, and Tieyan Liu. "On layer normalization in the transformer architecture." In International Conference on Machine Learning, pp. 10524-10533. PMLR, 2020.
>
> [3] Baevski, Alexei, and Michael Auli. "Adaptive Input Representations for Neural Language Modeling." In International Conference on Learning Representations. 2018.
>
> [4] Liu, Liyuan, Xiaodong Liu, Jianfeng Gao, Weizhu Chen, and Jiawei Han. "Understanding the Difficulty of Training Transformers." In Proceedings of the 2020 Conference on Empirical Methods in Natural Language Processing (EMNLP), pp. 5747-5763. 2020.

---

### Decision · Program_Chairs · 2021-09-27

**Decision:**

Accept (Poster)

**Comment:**

This paper studies the problem of training multilingual machine translation model in presence of imbalanced data, which is common in multilingual parallel corpus. The method is based on empirical observation that in multilingual MT training, "sharpness" of local curvature in the loss landscape causes interference among languages and instability. The paper then propose CATS, a curvature-based, online, per-language learning rate adjustment to improve the performance of low-resource languages in multilingual machine translation (MT) models.
Experiments in three scenarios  varying in multilinguality and data size (WMT-10, TED, OPUS-100) show that the technique works consistently, outperforming a widely-adopted temperature-based sampling baseline. Through a number of analysis and ablation experiments, they are able to provide insight into how and why the optimization technique works.
Experiments can be improved with more analysis (e.g. upsampling T). The paper also lacks comparison with some recent strong baselines for multilingual machine translation. The authors may consider moving some results and analysis from appendix to the main content. The overall writing, including figures, tables and captions, could be improved as suggested by all reviewers.